# Insulin Resistance in Obese Children: What Can Metabolomics and Adipokine Modelling Contribute?

**DOI:** 10.3390/nu12113310

**Published:** 2020-10-29

**Authors:** Francisco J. Rupérez, Gabriel Á. Martos-Moreno, David Chamoso-Sánchez, Coral Barbas, Jesús Argente

**Affiliations:** 1Centro de Metabolómica y Bioanálisis (CEMBIO), Facultad de Farmacia, Universidad San Pablo-CEU, CEU Universities, Urbanización Montepríncipe, Boadilla del Monte, 28660 Madrid, Spain; ruperez@ceu.es (F.J.R.); davidchamoso@gmail.com (D.C.-S.); 2Departments of Pediatrics & Pediatric Endocrinology, Hospital Infantil Universitario Niño Jesús, 28009 Madrid, Spain; gabrielangelmartos@yahoo.es; 3La Princesa Research Institute, 28006 Madrid, Spain; 4Department of Pediatrics, Universidad Autónoma de Madrid, 28006 Madrid, Spain; 5Centro de Investigación Biomédica en Red de Fisiopatología de la Obesidad y Nutriciόn (CIBEROBN), Instituto de Salud Carlos III, 28029 Madrid, Spain; 6IMDEA Food Institute, CEI UAM & CSIC, 28049 Madrid, Spain

**Keywords:** obesity, childhood, insulin resistance, adipokine, metabolomics

## Abstract

The evolution of obesity and its resulting comorbidities differs depending upon the age of the subject. The dramatic rise in childhood obesity has resulted in specific needs in defining obesity-associated entities with this disease. Indeed, even the definition of obesity differs for pediatric patients from that employed in adults. Regardless of age, one of the earliest metabolic complications observed in obesity involves perturbations in glucose metabolism that can eventually lead to type 2 diabetes. In children, the incidence of type 2 diabetes is infrequent compared to that observed in adults, even with the same degree of obesity. In contrast, insulin resistance is reported to be frequently observed in children and adolescents with obesity. As this condition can be prerequisite to further metabolic complications, identification of biological markers as predictive risk factors would be of tremendous clinical utility. Analysis of obesity-induced modifications of the adipokine profile has been one classic approach in the identification of biomarkers. Recent studies emphasize the utility of metabolomics in the analysis of metabolic characteristics in children with obesity with or without insulin resistance. These studies have been performed with targeted or untargeted approaches, employing different methodologies. This review summarizes some of the advances in this field while emphasizing the importance of the different techniques employed.

## 1. Introduction

In the 21st century new holistic approaches have been developed to tackle the systems biology challenge of discerning all the processes that characterize a living system at the molecular level. Genomics, proteomics, transcriptomics and more recently metabolomics have improved our understanding of what occurs in a biological system, as well as how it occurs. In the context of the study of clinical alterations, such as those that occur in obesity or insulin resistance, metabolomics can provide information about the actual metabolic phenotype in a given condition (alteration or illness), and how this metabolite set differs from a control (i.e., healthy) state.

Obesity, insulin resistance, metabolic syndrome, type 2 diabetes mellitus and many other metabolic alterations have been studied with different metabolomic approaches and technologies. However, most of these studies have been focused on adults. There is therefore a need to further our knowledge about how these alterations, which are at times hard to characterize in growing children, debut and how they develop in the pediatric population. Moreover, more biomarkers are needed to help in the adequate diagnosis of these conditions and for monitoring treatment efficiency and disease progression.

In light of our previous experience in the field, we have reviewed the current scientific literature on metabolomic studies in children and adolescents with obesity and/or insulin resistance. In order to provide insight into the type of results that can be obtained by using different metabolomics procedures, we have organized the available literature according to the methodology (i.e., untargeted, semi-targeted, targeted) used to perform the metabolomics study, highlighting the main results obtained with each approach. 

## 2. Childhood Obesity and the Development of Insulin Resistance

### 2.1. Childhood Obesity

The secondary complications of obesity are of major concern as quality of life is diminished and the mortality rate is increased. These concerns are amplified in the study of obesity in children. Not only has childhood obesity increased worldwide in recent years, but these cases are more severe and more precocious, resulting in the onset of health threatening complications at earlier ages [1,2]. Not only is there a dramatic need for programs/therapies to reduce the incidence of childhood obesity, but better diagnostic tools to identify those children at greater risk of developing severe complications are of utmost importance. One recurring problem in the approach to this problem is that observations in adult patients cannot always be directly applied to pediatric patients, especially prepubertal children.

Obesity is the excessive accumulation of adipose tissue that results in impairment of the patient’s physical and/or psychological function. Direct quantification of a patient’s body fat can be precisely performed by using methodologies such as bioimpedanciometry, dual X-ray densitometry, plethismography or hydrodensitometry. However, these advanced tools are not widely accessible in the usual clinical setting, setting aside investigation facilities or specialized clinical units. Consequently, the diagnosis of overweight or obesity is commonly established on the basis of an indirect estimation of patient’s body fat content by using body mass index (BMI = weight (kg)/(height (m)^2^), which has been shown to exhibit a good correlation with body fat content [3], although with some limitations (i.e., with extreme muscular mass development). In adult patients, 25 and 30 kg/m^2^ are widely accepted as the thresholds to diagnose overweight and obesity, respectively [4,5].

In children, agreement on a precise definition of obesity is more difficult than in adults. In children and adolescents, the BMI standardized for age and sex must be used, not its raw calculation. This has raised intense controversy regarding the establishment of “cut-off points” to define overweight and obesity and the population references that should be used [6]. In general, a child is considered to have excess body fat when their BMI is greater than the 95th percentile for his/her age and sex [3]. However, an optimum definition can be obtained by applying a cut-off point of BMI z-score above 2 compared with references from the same population, age and sex, thus meeting the proposal by the World Health Organization (WHO) [7]. Likewise, there is no consensus on the definition of morbid obesity in children and adolescents, with some authors suggesting a BMI z-score above 3 or 200% of the ideal body weight for height as possible cut-off points [8,9]. There is also no agreement on the definition of “early-onset” obesity, with ages below 5 or 2 years at the onset of the disease having been suggested by different authors [10]. This subgroup of patients with early-onset obesity is of particular interest because the excess of weight can be part of a syndrome or a monogenic disease. Another important difference between childhood and adult obesities is that in children, the development of the deleterious comorbidities associated with excess body fat may be a later event when considering the degree of obesity. This difference is in part due to the greater capacity of tissue turnover/regeneration in children.

### 2.2. Adipose Tissue in Obesity: The Importance of Age at Onset

The sequence of events in adipogenesis leading to the development of mature adipocytes from their embryonic pluripotential undifferentiated precursors requires their commitment towards the adipogenic lineage for the sequential formation of type I adipoblasts and after clonal expansion, type II preadipocytes. Growth arrest of type II preadipocytes will result in them becoming mature adipocytes and their accumulation of lipid droplets. In this cell linage the ability to synthesize and secrete adipokines is almost exclusively restricted to mature adipocytes [11,12,13].

As children continue to be in a period of growth and development, they possess a greater ability to adapt tissue morphology and function to their environment compared to adults. This is particularly relevant in the case of white adipose tissue, as its expandability is important in the metabolic complications in response to obesity [14]. Obesity results in histological, metabolic and endocrine changes in white adipose tissue [15]. These changes are determined by several factors: (1) The metabolic capacity of adipocytes to take-up free fatty acids (FFA) from the bloodstream, thus avoiding their ectopic deposition (lipotoxicity) [16]; (2) the production of chemoattracting proteins (chemokines) that results in an increase in specific proinflammatory populations of monocytes and macrophages [17,18,19] that substantially contribute to modifications in the adipokine secretion pattern of the tissue [20]; (3) the ability to recruit new adipocytes from preadipocytes, which has been postulated to occur once the former have reached a critical size [21] and (4) the change in the pattern of paracrine and endocrine adipokine secretion by hypertrophic adipocytes, as compared to normal-sized ones [22,23]. Each of these factors varies throughout the different stages of human development. 

In adults, either lean or obese, the adipocyte population in white adipose tissue remains relatively stable due to a balance between adipogenesis and apoptosis. In contrast, children and adolescents progressively increase the number of adipocytes in their adipose tissue, with this rise being even greater in obese compared to lean subjects due to a higher proliferation rate [24]. Consequently, early onset obesity is associated with an increase in adipocytes that could allow, at least transiently, for a limited degree of adipocyte hypertrophy and thus reducing the impact of obesity on the adipokine secretion profile and metabolic impairment at early ages, but conversely increasing the risk to develop severe obesity and metabolic comorbidities at later stages of life [3,24]. This is reminiscent of the “hyperplasic” model of obesity in children where there is an increased number of non-hypertrophic adipocytes versus a “hypertrophic” model of obesity in adults, with an increase in the volume of pre-existing adipocytes, resulting in a postulated difference in the impairment of the adipokine secretion profile between these two types of obesity [23,25].

### 2.3. Adipokines in Childhood Obesity

Leptin and adiponectin are the main adipokines involved in energy homeostasis and insulin sensitivity, respectively, among the extensive and continuously growing list of peptides secreted by adipose tissue. Additionally, a number of adipokines with proinflammatory actions (e.g., resistin, IL-6 and TNF-α, among many others) are produced in white adipose tissue, mainly by mononuclear stromal cells with the cellular composition of this tissue changing during the progression of obesity.

Leptin, found in the bloodstream both free and bound to the soluble isoform of its specific receptor, is mainly produced by mature adipocytes and acts as an adiposity signal. Leptin modulates the activity of several neuronal populations involved in the regulation of food intake and energy homeostasis in the central nervous system, including the proopiomelanocortin (POMC) producing neurons in the hypothalamic arcuate nucleus, exerting its main activity as a signal of energy sufficiency [26] with reported cases of severe human obesity due to leptin deficiency, reversible after recombinant leptin administration [27].

The circulating levels of leptin are directly correlated with body fat mass and adipocyte triglyceride content [28]. Gestational age and birth weight are the main determinant for its levels and bioavailability in the newborn [29]. Leptin levels significantly increase throughout pubertal development in females and decrease at the final stage of puberty in males. In contrast, circulating levels of leptin’s soluble receptor decrease in both sexes after pubertal onset, resulting in a puberty-related increase in free leptin that is more pronounced in adolescent females [30]. Obesity determines an increase in free leptin levels as a result of the increase in leptin and decrease of its soluble receptor, that is not reproduced in the spinal fluid, leading to “leptin resistance” [31].

Adiponectin is produced exclusively in mature adipocytes and circulates as polymers, with high molecular weight (HMW, 400–600 kDa) adiponectin postulated to be more metabolically relevant, particularly regarding its insulin sensitizing action [32]. This peptide acts through two specific receptors, adipoR1 and adipoR2, widely distributed but mainly located in muscle and liver, respectively. In muscle, liver and white adipose tissue adiponectin enhances insulin sensitivity and the promotion of fatty acid oxidation, with an increase indicating a beneficial apolipoprotein profile [33].

In newborns, adiponectin levels are higher than at later periods of life and positively correlate with gestational age and birth weight, with females having higher serum levels [29]. Postnatally serum adiponectin levels fall and its positive correlation with fat mass disappears around age 2 years, coincident with the increase in body fat [34]. In prepubertal children, most studies report a lack of differences between sexes, but males show lower adiponectin levels from mid-puberty onwards [30,32,33,34]. As opposed to leptin, serum adiponectin levels in adults are inversely related to the amount of body fat, with adult patients with obesity having decreased circulating adiponectin levels [35]. However, this inverse correlation between body fat and serum adiponectin levels is not present in all patients and is influenced by adipocyte size [36]. In adolescents with obesity, an inverse correlation between adiponectin levels, body fat and insulin resistance, similar to that reported for adults, has been demonstrated [34,37].

### 2.4. Carbohydrate Metabolism Impairment in Childhood Obesity: Insulin Resistance

As stated above, overt metabolic impairment in children with obesity can be delayed due, at least in part, to the singularities of young adipose tissue. This is particularly evident regarding carbohydrate metabolism. Although there is wide geographic and ethnic variability, we recently demonstrated a minimal incidence of type 2 diabetes mellitus (T2DM) in children despite the high number of patients affected with severe obesity in our country [38]. In contrast, the prevalence of initial glycemic alterations (defined as “prediabetic conditions”) and more importantly, the number of patients showing peripheral resistance to insulin induced glucose caption (“insulin resistance” [IR]) is much higher [38].

However, the definition of IR in the clinical setting is extremely controversial, particularly in pediatrics, and continues to be a matter of intense debate in the international community. Although the clamp tests used in investigational facilities are considered the “gold standard” for IR validation, common clinical determinations can be used for the calculation of IR indexes on the basis of fasting (HOMA-IR index) and/or postprandial (insulinogenic and insulin sensitivity indexes) measurement of glycaemia and insulinemia [39]. However, these indexes have been shown to correlate well with those derived from studies using euglycemic-hyperinsulinemic clamps [40].

In the first definition of “X” or metabolic syndrome, IR was suggested as the pathophysiological basis of the remaining obesity-associated metabolic derangements [41]. The definition of metabolic syndrome has been subsequently modified, conferring a primary role to the presence of abdominal obesity and in particular visceral adipose tissue, with waist circumference showing a better association to cardiovascular risk than BMI itself [42]. This observation has been extended also to the pediatric and adolescent population and, consequently, abdominal circumference and not BMI has been considered as the anthropometric criterion for the definition of metabolic syndrome in children above 10 years of age [43] Surprisingly, in children the criteria to define metabolic syndrome regarding carbohydrate metabolism only take into consideration the presence of impaired fasting glucose (IFG) and/or T2DM, whereas IR is not considered [43]. Similarly, hyperinsulinemia/IR is not usually considered in the definition of “metabolically healthy” obesity in childhood [44], although some authors point out its relevance [45,46]. These consensus statements and criteria, originating mainly from studies in the adult population, should be revised for children and adolescents. Indeed, glycemic alterations are late, or frequently absent, findings in children with obesity in our environment, whereas a rise in both fasting and postprandial insulinemia can be identified as the initial steps of carbohydrate metabolism impairment, particularly in young children with obesity [38].

Additionally, a bidirectional influence between leptin and insulin exists, with hyperinsulinemia enhancing leptin production and increased free leptin levels increasing insulin resistance, with “leptin resistance” and IR usually coexisting in obesity, even at young ages [47]. The development of these hormonal derangements is gradual and identification of biomarkers to precociously predict the risk of developing serious metabolic complications is of great importance. Consequently, the application of new techniques, such as metabolomics, in combination with adipokine measurements can afford additional information to address the pathophysiological relevance of this controversial condition of IR in childhood, even when its definition is based upon analytical criteria derived from usual clinical practice.

## 3. Adipokine Modelling

As stated above, obesity-induced modifications in adipose tissue differ between children and adults and there is insufficient information regarding how these changes affect the development of further complications and metabolic syndrome in pediatric patients with obesity. 

We have recently reported a novel approach to identifying biomarkers of insulin resistance in children, both prepubertal [48] and pubertal [49], with obesity. We found that combined sets of cytokines, adipokines and chemokines can be used as models to predict insulin resistance. In both pediatric age groups specific factors, including tumor necrosis factor (TNF)α, eotaxin, insulin-like growth factor (IGF)-1, leptin, triglycerides (TGL), monocyte chemeoattractant protein (MCP)1 and brain derived neurotrophic factor (BDNF), were identified as biomarkers involved in insulin resistance. Interestingly, in adolescents with obesity the presence of insulin resistance is influenced by the chemokines MCP-1 and eotaxin, as well as the growth factor platelet derived growth factor (PDGF)-BB, in a sex independent manner. These three biomarkers are part of the main component that together with stromal cell derived factor (SDF)1α and BDNF, determine 27.7% of the variance associated with insulin resistance. In prepubertal obese children, we defined two predictive models that include the combination of leptin, TG/HDL, IGF-1, TNFα, MCP1 and PDGF-BB with an optimal sensitivity and specificity of 93.2%. Hence, we suggest that the combination of these circulating parameters from a single fasting sample could be useful to predict insulin resistance in prepubertal children with obesity. These adipokines in combination with other biomarkers, such as specific metabolites, could possibly serve as an even more powerful predictive model. 

## 4. Metabolomics

Metabolomics, in which potentially all small-molecule metabolites (the metabolome) are identified and at some level quantified, is generally acknowledged to be the omics discipline that supplies the most rapid and clearest information about the phenotype. For this reason, it is greatly appreciated for its role in biomarker discovery. The rationale underlying “the study of the metabolome” (i.e., metabolomics) is based on the assumption that the metabolome is the reflection of all the processes that might be occurring at one moment (time-course changes) or be altered under one condition (changes due to disease, treatment, etc.). The information for such study can be gathered through different experimental approaches that receive different names.

Unfortunately, in the field of metabolomics the terminology is not yet fully standardized, and clear unequivocal terms have not been assigned to each methodology. This lack of standardization might lead to confusion, as the same term may be used for different types of studies, whereas the same type of information might be obtained by using similar methodologies that have received different names. For clarity we have classified the studies as untargeted, targeted and semi-targeted, although the limits are sometimes diffuse, and methodologies are very often not accurately described as to establish a clear classification. Metabolomics studies related to obesity and/or insulin resistance in non-adult subjects are shown in Table 1. Analysis of the studies performed so far indicate that they include a limited number of individuals, compare different experimental groups (obese, normal weight, different ages and racial origin) with the only constant of IR and therefore, it is not surprising that results are also heterogeneous.

### 4.1. Untargeted Metabolomics

Untargeted metabolomics is an approach that highly contrasts with targeted analysis (the classical way to study metabolism), where a limited number of specific known compounds are analyzed. The rationale to employ an untargeted approach is that modern spectrometric/spectroscopic techniques can generate a huge amount of information in the form of signals coming from all the metabolites in the samples. Such signals can be statistically compared between case/control groups to identify the characteristics that differentiate these conditions without a priori hypothesis. Once isolated the specific spectral characteristics are associated with metabolites (as “identified” or more precisely, “annotated”). The analytical workflow in untargeted metabolomics is still far from being exhaustively protocolized, but it can be described as a sequential process that starts with the biological question, and then follows a series of steps of experimental design, sample treatment, instrumental analysis, signal processing, data pretreatment, statistical analysis, metabolite annotation, validation and biological interpretation [50]. This untargeted approach is of great interest in what can be called the “discovery phase” resulting in the possibility to unveil new, not previously described compounds that can be related to the characteristic of interest, for example metabolic alterations of insulin resistance in the context of childhood obesity.

To gather information about the metabolome, the instrumental analysis is generally conducted through either nuclear magnetic resonance (NMR) or mass spectrometry (MS). In recent years MS has become the most employed technique in metabolomics [50]. It can be hyphenated to a separation technique (gas chromatography, GC; liquid chromatography, LC; capillary electrophoresis, CE; or supercritical fluid chromatography, SFE) or not (Direct MS, DMS). MS benefits from detection permitted at high sensitivity and structural elucidation based on spectral libraries and tandem MS, even in complex biological samples. Nevertheless, as the diversity of chemical characteristics of the metabolites is so broad, there is no single technique that can cover the full range of metabolites in a sample. For this reason, the results obtained are strongly dependent on the technique and the methodology that have been used in the analysis and obtaining really non-biased results requires a “multiplatform” approach [51,52].

GC-MS is usually combined with LC-MS [51,53,54,55,56] as the biochemical information is complementary [57]: GC-MS is very well suited for the analysis of metabolites related to central carbon metabolism such as short chain organic acids, amino acids, monosaccharides, fatty acids, disaccharides and cholesterol, and LC-MS is adequate for less polar molecules, i.e., lipids [58]. GC-MS can even stand alone, and provide information of a large set of compounds, even of a single family such as 75 steroid-related metabolites in the context of childhood obesity [59]. However, the most widely employed technique for metabolic fingerprinting is reversed-phase LC-MS, which involves the minimum requirement for sample treatment and alteration or hydrolysis of the metabolites during the analysis among the hyphenated techniques [50]. CE-MS shows clear benefits for multiplatform untargeted metabolomics [60], and the analysis of amino acid-related compounds. Mastrangelo et al. [51] found differences in amino acids, acylcarnitines, polyamines and xanthines, among the metabolites that could be measured with this technique, as part of a multiplatform approach. The NMR profile can contain qualitative and quantitative information on hundreds of different small molecules present in the sample. Although sensitivity is poorer in NMR than in MS, the robustness and elucidation capabilities are usually claimed as its main advantages. 

Currently, the main bottleneck in metabolomics is the identification of the metabolites of interest when they are found in LC-MS, CE-MS or DMS measurements. The identification process starts with the quest for information in publicly available databases, with the input being the exact mass and the output a chemical identity. However, databases for metabolomics are not fully standardized and the different databases vary in the number of records (compounds), in the fields for each record (compound properties), as well as the searchable fields (mass, m/z, MS/MS spectrum, name, etc.). To facilitate the simultaneous query in different databases, tools such as CEU Mass Mediator (http://ceumass.eps.uspceu.es/) allow one to obtain the information from the available databases, together with other utilities [61]. Given the fact that the discovery phase in untargeted metabolomics is usually performed measuring thousands of signals in a limited set of samples, validation of results with a different analytical technique and in a different cohort strongly increases their reliability.

### 4.2. Untargeted Metabolomics Applied to Obesity and Insulin Resistance in Children and Adolescents

Researchers have applied different untargeted approaches in the study of obesity and IR. The broadest metabolite coverage when studying the effect of obesity and IR was obtained by our group by using three analytical techniques (LC-MS, GC-MS and CE-MS) [51]. Amino acids, gut microbiota by-products and lipids were found altered, and the study also highlighted that these modifications were sex specific, with differences in boys and girls even though the children in these studies were prepubertal. The most relevant findings were later validated with a target method in a bigger cohort [62], which showed an increase in branched chain amino acids (BCAA) and aromatic amino acids (Phe, Tyr and Trp), with the most altered pathways being the urea cycle, alanine metabolism and the glucose-alanine cycle.

The robustness and capability to perform studies with large sample number should be one of the potential advantages of NMR, although it has been applied only to small studies: Tricó et al. [63] showed the relevance of specific patterns of amino acids and carbohydrates to predict future (2.3 years) worsening in glycemia, whereas Hosking et al. [64] demonstrated that these types of metabolites were different between boys and girls, and insulin resistance was worse in girls than in boys. Both studies were performed in normal weight adolescents.

### 4.3. Semi-Targeted Metabolomics

Advances in instrumentation and software processing tools, together with massive data storage capabilities have permitted the development of what we could call semi-targeted analysis: A non-biased sample treatment and generic chromatographic conditions, coupled to an MS device that obtains fragmentation spectra of each and every compound that can be detected. Such spectra are compared with those included in a database built with the analysis of real standards [65]. This methodology increases the throughput of the process, because only those compounds previously included in the database are sought in the samples. This reduces the time devoted to the elucidation of the unknown compounds. 

### 4.4. Semi-Targeted Metabolomics Applied to Obesity and Insulin Resistance in Children and Adolescents

One of the drawbacks of untargeted MS metabolomics approaches is that, due to inherent variability of the signals from sample to sample, the discovery phase studies are usually performed in small cohorts, usually less than 125 individuals per group, thus less than 250 for the entire study (See Table 1). Nevertheless, with the most commonly cited semi-targeted approach with LC-MS/MS combined with GC-MS [66], samples of over 700 Hispanic children were successfully compared (obese/non obese, boys/girls) [56]. Moreover, the use of such common methodology applied to four different untargeted studies permitted the performance of an individual participant meta-analysis about obesity and insulin resistance in children: one sphingomyelin (SM(d18:2/14:0)) was positively associated with obesity, whereas association with HOMA was found for alanine (positive) and acylcarnitines and non-esterified fatty acids (negative) [67]. Perng et al. applied such data-driven LC-MS/MS approach combined with GC-MS [66] to find differences in a set of more than 3000 compounds in studies related to obesity [53] and metabolic risk [55], and they also found that the most prominent changes were related to branched chain amino acids (BCAA) and acylcarnitines, although not always with a consistent trend, because BCAA was not associated with worsening metabolic health during early adolescence and the relationship of BCAA with fasting glucose or serum triglycerides was different in boys and girls [55].

### 4.5. Targeted Metabolomics

While untargeted metabolomics is the choice for the discovery of new previously unknown compounds, the capabilities of available analytical instrumentation also allows several hundreds of well-characterized compounds to be measured simultaneously. However, when the systems are programmed to measure one set of metabolites, there will be no signal from other compounds. This approach, very popular in clinical studies, is considered as targeted analysis. By measuring a large set of metabolites with this targeted approach researchers can perform “targeted metabolomics.” But metabolomics does not mean just measuring a large number of metabolites, and this denomination alone is far from being a clear indication of the methodology employed in its analytical determination.

In this type of studies, the keyword “metabolomics” seems to indicate only that a large set of metabolites has been simultaneously determined, but this is far from being a clear indication of the methodology employed in its analytical determination. 

These studies are called “metabolomics” because they generate a multivariate space, with all the metabolites that can be measured. This has become useful to find mathematical associations between metabolites, which can help to define possible single theragnostic biomarkers such as asymmetric dimethyl arginine (ADMA), which was found to be associated to insulin resistance in adolescents [68]. Moreover, the possibility of identifying a set of metabolites that together show predictive power is one of the biggest achievements of the multivariate metabolomic approach. Such strategy has resulted in the proposal of the so-called metabolic signature [69] (later called BCAA-related signature [70]), metabolomic signature [71], metabolite profiling [72], metabolomic profile [53,73,74,75,76,77,78,79], or metabolic phenotype [80,81] and although the names differ, they share the same underlying concept.

Something common to all metabolomics studies, whether untargeted or targeted, is the application of multivariate statistical analysis, both unsupervised (e.g., principal component analysis, PCA) and supervised (e.g., projection on latent structures/partial least squares-discriminate analysis, PLS-DA). Ideally, this should lead to the proposal of strong biomarkers of insulin resistance such as different adipokines [82]. Nevertheless, and despite the accumulated evidence [83], biomarkers from metabolomics studies such as BCAA, aromatic amino acids, acylcarnitines, or some lipids (Table 1) are not measured in the routine of the clinics of obesity or insulin resistance.

### 4.6. Targeted Metabolomics Applied to Obesity and Insulin Resistance in Children and Adolescents

In the field of obesity and insulin resistance, the concentrations of BCAA and acylcarnitines have been extensively studied by using targeted metabolomics since Newgard et al. described “a BCAA metabolic signature” associated to obesity and insulin resistance [69]. In recent years, several studies have been performed to gain more evidence concerning the relationship between obesity and BCAA. In adults, most of these studies have consistently shown the association of obesity, insulin resistance and type 2 diabetes with elevated BCAA, aromatic amino acids, C3 and C5 acylcarnitines and glutamate and alanine [70], and BCAA have been proposed as good biomarkers of obesity and insulin resistance in adult individuals [83]. Along with BCAA, an alteration in the levels of acylcarnitines could also be used to discriminate between children with or without insulin resistance [51,56,62,67,76,84,85].

However, its usefulness as a biomarker in childhood obesity and insulin resistance remains to be elucidated. As previously mentioned, the studies shown in Table 1 indicate some of the difficulties for performing meta-analysis with them, as not only do they present differences in terms of the methodology used to carry out the instrumental analyses, but the size, origin and characteristics of the cohorts are also different. These differences in the design of the studies might justify some apparent discrepancies: In most of the studies, BCAA is increased in obese children and adolescents [53,56,72,76,85]. In addition, prepubertal children with obesity and insulin resistance present an increase in BCAA compared to obese prepubertal children without insulin resistance [51,62]. The same trend was shown in adolescents [63]. However, other studies have shown no alteration or even a decrease in BCAA levels between obese children as compared to children with normal weight [55,64,79,84]. This implies that future studies must be carried out to clearly elucidate the association between BCAAs and insulin resistance in non-adult populations. Such discrepancies are not related to differences in the methodology used to gather the information about metabolites, despite that there is no uniformity in the terminology. Moreover, as stated above it is important to differentiate metabolomic profiles between boys and girls during childhood. A study by Newbern et al. [74] reported an increase in BCAA levels and BCAA by-products in boys compared to girls, together with an inverse relationship between adiponectin and BCAA in boys. 

Most of the targeted metabolomics studies concerning insulin resistance and obesity in childhood have focused on amino acids and acylcarnitines. Nevertheless, measurement of the alterations in the lipid profile with a metabolomics approach, i.e., lipidomics, has demonstrated a strong correlation of one lysophosphosphatidylcholine (LPC(14:1)) and one phosphatidylcholine (PC(16:0/2:0)) with cardiovascular disease risk factors in adolescents [86]. In addition, alterations in steroid hormone levels have been found in children with insulin resistance [53,56,59,76].

### 4.7. Combining Metabolomics Information in Obesity and Insulin Resistance

As no single technique can provide coverage of the whole metabolite, the samples must be analyzed by different techniques, and the information must be integrated. Metabolomics can supply a large amount of useful information, but other determinations are still necessary. For instance, Newgard et al. combined information of the so-called “conventional” metabolite determination (glucose, lactate, cholesterol, etc.) with the targeted MS/MS analysis of acylcarnitines and amino acids, plus the free and total fatty acids, and short-chain organic acids by GC/MS to characterize the metabolic signature that was different between lean and obese [69].

We analyzed possible correlations between the metabolites measured and other clinical determinations such as the HOMA index, total triglycerides, leptin and adiponectin [57]. In the ROC analysis, the combination of leptin and alanine showed a high IR discrimination value in the whole cohort (area under curve, AUCALL = 0.87), as well as in boys (AUCM = 0.84) and girls (AUCF = 0.91) when considered separately. However, the specific metabolite/adipokine combinations with highest sensitivity were different between the sexes. Therefore, combined sets of metabolic, adipokine and metabolomic parameters can identify pathophysiological relevant IR in a single fasting sample, suggesting a potential application of metabolomic analysis in clinical practice to better identify children at risk without using invasive protocols.

Based on our current understanding of this problem, more research is clearly needed to elucidate reliable biomarkers for future complications in childhood obesity, including employing different types of samples, such as feces. Diseases associated with lifestyle, as well as their complications, are complex and multifactorial in nature. Genetic heritage, dietary habits, and other environmental factors, as well as their interaction with the microbiome, conditioning gene expression and transcription and the subsequent regulation of protein translation and activity, all impact on the metabolic outcome. All these factors are molecularly related to the metabolome. Moreover, the role of factors such as the gut microbiome and low-grade inflammation in modulating the response to insulin and other hormones cannot be questioned but is exceedingly difficult to quantify. 

**Table 1 nutrients-12-03310-t001:** Metabolomic studies about obesity, insulin resistance or type 2 diabetes mellitus (T2DM) in children.

Methodology	Instrumental Analysis	Disease	Study Design	Sample	Findings	Ref.
Untargeted	LC-MS, CE-MS, GC-MS	Obesity and IR	Fingerprinting study: 60 prepubertal obese children. Boys (*n* = 30, 50% IR and 50% non-IR) Girls (*n* = 30, 50% IR and 50% non-IR) Validation study: 100 prepubertal obese children. Boys (*n* = 50, 50% IR and 50% non-IR) Girls (*n* = 50, 50% IR and 50% non-IR)	Serum	IR vs non-IR:	[51]
					Inflammation, central carbon metabolism and gut microbiota are the most altered processes. Increased BCAA, ArAAs, Ala, Pro, Pyr, taurodeoxycholate, glycodeoxycholate, piperidine, pyroglutamate. In females, increased free carnitine, propionylcarnitine and butyrylcarnitine, but in males only propionylcarnitine.	
Untargeted	LC-MS/MS	Metabolic Risk	Boys (*n* = 113) Girls (*n* = 125) (8–14 years)	Serum	Metabolic Risk: In girls:	[54]
					Positive association of DG(16:0/16:0), 1,3-dielaidin, myo-inositol, and urate. Inverse association of thymine, dodecenedioic acid, and *n*-acetylglycine with metabolic risks.	
					In boys::	
					Positive associations of BCAA, DG(16:0/16:0), tyrosine, and 5′-methylthioadenosine.	
Untargeted	NMR	IR	Cross sectional study: 78 non diabetic adolescents (8–18 years) Longitudinal study: 16 subjects after a mean follow-up of 2.3 years	Plasma	Higher baseline 2-hydroxybutyrate and BCAA levels in insulin resistant youth and predict worsening of glycemic control Alterations of 2-hydroxybutyrate metabolism predict incipient deterioration of β-cell function and longitudinal worsening of glycemic tolerance.	[63]
Untargeted	NMR	IR	170 healthy normal weight children (5, 14 and 16 years)	Serum	IR higher in girls than in boys. In healthy normal weight children IR was associated with reduced concentrations of BCAA, 2-ketobutyrate, citrate and 3-hydroxybutyrate, and higher concentrations of lactate and alanine.	[64]
Semi-targeted	LC-MS/MS, GC-MS	Obesity	Obese (*n* = 84) Overweight (*n* = 28) Normal-weight (*n* = 150) Median age 7.7 years 50% boys 50% girls	Plasma	OB vs NW:	[53]
					Increased BCAA (Val, Leu, Ile) and androgen hormones (DHEA-S).	
Semi-targeted	LC-MS/MS, GC-MS	Obesity and IR	Hispanic children Obese (*n* = 450) Non-obese (*n* = 353) Boys (*n*= 405) Girls (*n* = 398). (4–19 years). Mean age 11.1 years	Plasma	OB vs NOB:	[56]
					Increased BCAA and acylcarnitine catabolism and changes in nucleotides, lysolipids, steroid derivatives and inflammation markers. Reduced fatty acid catabolism. BCAAs, ArAAs, aspartate, dipeptides, citrate, asparagine, glycine and serine is associated with risk factors for IR, hyperleptinemia, hypertriglyceridemia, hyperuricemia and inflammation.	
Semi-targeted	LC-MS/MS, GC-MS	Obesity	Longitudinal study for 5 years: Obese (*n* = 68) Overweight (*n* = 23) Normal weight (*n* = 122) 48.8% boys Median age 7.7 years	Plasma	BCAA is not associated with worsening metabolic health during early adolescence. Inverse association of the BCAA pattern with a change in fasting glucose in boys. Direct relation of BCAA pattern with a change in serum triglycerides in girls. Higher score for androgen hormone pattern at baseline corresponds with a decrease in leptin an increase in CRP in girls.	[55]
Targeted	LC-MS/MS, GC-MS	Obesity and IR	100 prepubertal obese children. Boys (*n* = 50, 50% IR and 50% non-IR) Girls (*n* = 50, 50% IR and 50% non-IR) 5–10 years	Serum	IR vs non-IR:	[62]
					Higher ALT, GPT and TAG levels Higher leptin and reduce leptin/adiponectin ratio Increase BCAA, ArAAs (Phe, Tyr and Trp), and Ala The most altered pathway is the urea cycle, alanine metabolism and the glucose-alanine cycle. C12 acylcarnitine and methionine correlate with HOMA-IR exclusively in males	
Targeted	MS/MS	Obesity and T2D	Case-control: Obese (*n* = 64) Obese with TD2 (*n* = 17) Normal-weight (*n* = 39) 12–17 years	Plasma	T2D vs OB/NW:	[87]
					Decreased BCAA.	
					T2D/OB vs NW:	
					No difference in long-chain AcylCN Reduced short and medium-chain AcycCN No defects in fatty acid or amino acid metabolism	
					No differences in fasting FFA levels	
Targeted	MS/MS	Obesity, IR and T2D	Case-control: Obese (*n* = 57) Obese prediabetes (*n* = 27) Obese T2D (*n* = 17) Normal-weight (*n* = 38) 13–14 years	Plasma	BCAA and BCAA intermediates correlated: positively with insulin sensitivity and DI	[79]
Targeted	LC-MS/MS	Obesity and IR	Cross sectional study: 69 healthy children and adolescents 8-18 years Longitudinal cohort study in subset: Subgroup of 17 participants 8-13 years	Plasma	OB vs NW:	[72]
					Increased BCAA Increased BCAA not associated with measures of insulin resistance at baseline. Baseline BCAAs predicted HOMA-IR at 18 months. Elevations in the concentrations of BCAAs were associated with reduced insulin sensitivity at 12 months.	
Targeted	MS/MS	Obesity and IR	Cross-sectional study: Obese (*n* = 82) Boys (*n* = 41) Girls (*n* = 41) 12–18 years	Plasma	BCAA levels and by products of BCAA catabolism are higher in males than females with similar BMI. In males, HOMA-IR correlated:	[74]
					Positively: BMI z-score, BCAA, uric acid, long-chain acyl-carnitines Negatively: fatty-acid oxidation products	
					In females, HOMA-IR correlated:	
					Positively: BMI z-score	
					Adiponectin correlated inversely with BCAA and uric acid in males, but not females	
Targeted	LC-MS/MS	Obesity and IR	Identify biomarkers predictive of future disease risk- Obese (*n* = 46) Obese to normal weight (*n* = 18) Normal-weight (*n* = 45) 9–11 years	Plasma	Baseline BCAA concentration as a predictor of future risk of insulin resistance and metabolic syndrome OB vs NW:	[85]
					Increased levels of BCAA, Tyr, Phe, 2-AAA and several acyl-carnitines Lower levels of acyl-alkyl phosphatidylcholines	
Targeted	MS/MS	Obesity and IR	Longitudinal study: 80 obese Caucasian children. 40 participate in one-year lifestyle interventions 8–15 years	Serum	Tyr was the only metabolite significantly associated with HOMA-IR at baseline and after 1-year intervention. No association between HOMA-IR and BCAA.	[84]
Targeted	MS/MS	Obesity and IR	430 control (13–15 years). 91 morbid obese (12–16 years)	Plasma	Accumulation of ADMA is associated with modulation of insulin signaling and insulin resistance. ADMA decreased after obesity intervention program	[68]
Targeted	MS/MS, LC-MS/MS	Obesity and IR	Meta-analysis 1020 pre-pubertal children from three European studies. 8–10 years	Plasma	Positive association of SM (32:2) with BMI z-score. SM 32:2 as a potential molecular marker for mechanistic alterations involved in the pathogenesis of obesity. Ala and Tyr was associated positively with HOMA-IR. Acylcarnitines and non-esterified fatty acids were negatively associated with HOMA.	[67]
Targeted	GC-MS	Obesity and IR	20 obese with IR 67 obese without IR 8.5–17.9 years	Urine	The steroidal signature IR vs non-IR:	[59]
					High adrenal androgens, glucocorticoids and mineralocorticoid metabolites Higher 5α-reductase and 21-hidroxylase activity Lower 11bHSD1 activity	
					The authors suggest a vicious cycle model, whereby glucocorticoids induce IR.	
Targeted	MS/MS	Obesity and Metabolic Risk	Non-OW/OB and low MetRisk (*n* = 335) Non-OW/OB and high MetRisk (*n* = 29) OW/OB and low MetRisk (*n* = 58) OW/OB and high MetRisk (*n* = 102) Girls 48.3% Boys 51.7% 11–16 years	Plasma	Lower levels of LCFA in non-OW/OB with high MetRisk and OW/OB with high MetRisk compared to mon-OW/OB with low MetRisk. Higher levels of BCAA metabolite pattern in OW/OB with high MetRisk compared to non-OW/OB with low MetRisk. Higher levels of DAG in OW/OB with high MetRisk vs non-OB/OW with low MetRisk. Higher score of androgen steroid hormones pattern in OW/OB with high MetRisk compared to Non-OW/OB with low MetRisk. Higher levels of AcylCN in non-OW/OB with high MetRisk compared to non-OW/OB with low MetRisk. Lower levels of AcylCN in OW/OB with high MetRisk compared to Non-OW/OB with low MetRisk.	[76]

Abbreviations: 11bHSD1: 11β-hydroxysteroid dehydrogenase type 1; 2-AAA: alpha amino adipic acid; AcylCN: acylcarnitines; ADMA: asymmetric dimethylarginine; Ala: alanine; ALT: alanine transaminase; ArAAs: aromatic amino acids; BCAA: branched chain amino acids; BMI: body mass index; CE-MS: capillary electrophoresis – mass spectrometry; CRP: C-reactive protein; DAG: diacylglycerides; DG: diglyceride; DI: disposition index; GC-MS: gas chromatography – mass spectrometry; GPT: gamma-glutamyltransferase; HOMA-IR: homeostatic model assessment – insulin resistance; IR: insulin resistance; LCFA: long-chain fatty acids; LC-MS: liquid chromatography – mass spectrometry; NMR: nuclear magnetic resonance; NOB: non obese; NW: normal weight; OB: obese; OW: overweight; Phe: phenylalanine; Pro: proline; Pyr: pyruvate; SM: sphingomyelin; T2D: type 2 diabetes; TAG: triacylglycerides; Trp: tryptophan; Tyr: tyrosine.

## 5. Conclusions

Not only is the concept of obesity in children and adolescents unclear, but the definition of insulin resistance continues to be controversial. In this regard, the combined analysis of adipokines (particularly leptin and adiponectin), growth factors, inflammatory markers, chemokines, metabolic and metabolomic markers could be useful to predict the existence of insulin resistance in children with obesity prior to overt glucose metabolism impairment.

The evolution of obesity and its comorbidities differ between children and adults and more studies are necessary in children to define insulin resistance, as well as metabolic syndrome, and determine its implications in further complications.

New and precise markers of the evolution of glucose metabolism in children and adolescents with obesity are necessary to provide a correct diagnosis and early intervention.

Metabolomics, untargeted, targeted and the combination of these, is a powerful new technology to understand metabolism and to highlight possible biomarkers with clinical relevance.

Metabolomics can provide valuable information from bench to bedside and backward, and the information gathered from large metabolomics studies can be applied to the pursuance of precision nutrition. Ideally, we will be able to relate the presence of some metabolites, at least to some extent, to characterize the individual needs in terms of nutrition.

More studies are clearly necessary to precisely determine the progression of alterations in glucose metabolism in young patients with obesity to identify clear biomarkers of risk of further complications. The metabolites that are most often found associated with obesity and/or insulin resistance (BCAA and acylcarnitines) still need to be studied in children and adolescents. Moreover, other biomarkers coming from untargeted studies (related to inflammation or the gut microflora) should be tested in the clinic.

The patient’s sex must be taken into consideration even in prepubertal periods.

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
