# Peer review of "Insulin Resistance in Obese Children: What Can Metabolomics and Adipokine Modelling Contribute?"

_nutrients, 2020, doi:10.3390/nu12113310_

Round 1

Reviewer 1 Report

Excellent contribution. However, since this paper reviews the existing literature, the degree of self citations is a bit too much. Would it please be possible to omit some of the self citations and replace with references from other groups ?

Author Response

We greatly appreciate the reviewer´s consideration. Following this recommendation, we have removed 6 references to our previous studies, and we have included additional references as requested.

Reviewer 2 Report

The article of Ruperez et al. is an interesting review that addresses the contribution of metabolomics to the identification of predictive biomarkers for the development of metabolic complications in obese children and adolescents. There are nevertheless a few points that need to be clarified or completed.

Globally, an effort to restructure this review is necessary, as it is difficult to read in its present form.

A introduction is missing and should introduce the review.

The language is sometimes familiar “we speak”, in “our mean “, “we could call” …

Some references should be added (sentences: L41-L43; L56-57; L65-67; L67-69; L151-153

Please indicate the limitations you mentioned (L53-56).

It is clear nowadays that BMI alone is not sufficient to properly evaluate the cardiometabolic risk associated with increased adiposity and that waist circumference measurement is a better predictor of health outcome in obese patients. What about children and adolescents? This topic deserves to be discussed.

L281; L306; L352; L383: It should be stated in the title of these paragraphs "in children and adolescents".

Targeted metabolomics: please explain more clearly what targeted metabolomics is, as done for untargeted and semi-targeted metabolomics.

L328-333 and L344-351: these sentences should be moved to the paragraph dealing with obesity and insulin resistance.

Conclusion: From all the literature analysis performed on metabolomics in children and adolescents, does a common signature emerge or what are the possible directions? It would be necessary to conclude on this point. Moreover, there is no mention of the contribution of adipokine modelling in the conclusion, this point is missing, as the title raises the question “what can metabolomics and adipokine modelling contribute?”.

Author Response

The article of Ruperez et al. is an interesting review that addresses the contribution of metabolomics to the identification of predictive biomarkers for the development of metabolic complications in obese children and adolescents. There are nevertheless a few points that need to be clarified or completed.

We thank the reviewer for this positive feedback, and the points that have allowed us to improve the manuscript.

Globally, an effort to restructure this review is necessary, as it is difficult to read in its present form.

We have tried to rearrange some parts as requested, clarifying some points (see below). We hope that this is sufficient so that it is now less difficult to read, as the reviewer did not make specific comments as to how to restructure the manuscript.

A introduction is missing and should introduce the review.

We have included a short introduction as suggested.

The language is sometimes familiar “we speak”, in “our mean “, “we could call” …

We have revised the manuscript and replaced this usage where necessary. Nevertheless, 1st person statements are now thought to be adequate, and are even encouraged, in a critical review like this, as it indicates a stronger involvement of the authors in the writing. See for instance by David H. Foster, in the OUP blog: https://blog.oup.com/2018/01/first-person-pronouns-passive-voice-scientific-writing/

Some references should be added (sentences: L41-L43; L56-57; L65-67; L67-69; L151-153

We have included references as suggested.

Please indicate the limitations you mentioned (L53-56).

These limitations are now included.

It is clear nowadays that BMI alone is not sufficient to properly evaluate the cardiometabolic risk associated with increased adiposity and that waist circumference measurement is a better predictor of health outcome in obese patients. What about children and adolescents? This topic deserves to be discussed.

The following information has been included as requested:

The definition of metabolic syndrome has been subsequently modified, conferring a primary role to the presence of abdominal obesity and, particularly, visceral adipose tissue, with waist circumference showing a better association to cardiovascular risk than BMI itself [42]. This observation has been extended also to the pediatric and adolescent population and, consequently abdominal circumference and not BMI has been considered as the anthropometric criterion for the definition of metabolic syndrome in children above 10 years of age [43])

L281; L306; L352; L383: It should be stated in the title of these paragraphs "in children and adolescents".

Following the reviewer´s recommendation, we have included "in children and adolescents" in the subtitles.

Targeted metabolomics: please explain more clearly what targeted metabolomics is, as done for untargeted and semi-targeted metabolomics.

We have modified the paragraph according to the suggestion:

This approach, very popular in clinical studies, is considered as targeted analysis. By measuring a large set of metabolites with such targeted approach researchers can perform “targeted metabolomics”. But metabolomics does not mean just measuring a large number of metabolites, and this denomination alone is far from being a clear indication of the methodology employed in its analytical determination.

L328-333 and L344-351: these sentences should be moved to the paragraph dealing with obesity and insulin resistance.

 Following the reviewer’s advice, we have moved L328-333 to the mentioned paragraph. However, we think that the text in L344-351 corresponds to general statements regarding biomarkers, and that they are not focused specifically on insulin resistance. Hence, we hope the reviewer understands why we have left them in their original position.

Conclusion: From all the literature analysis performed on metabolomics in children and adolescents, does a common signature emerge or what are the possible directions? It would be necessary to conclude on this point.

We have included in the conclusions:

(…) and to highlight possible biomarkers with clinical relevance.

(…) The metabolites that are most often found associated with obesity and/or insulin resistance (BCAA and acylcarnitines) still need to be studied in children and adolescents. Moreover, other biomarkers coming from untargeted studies (related to inflammation or the gut microflora) should also be tested in the clinic.

Moreover, there is no mention of the contribution of adipokine modelling in the conclusion, this point is missing, as the title raises the question “what can metabolomics and adipokine modelling contribute?”.

Thank you for pointing this out. We have added the following statement:

In this regard, the combined analysis of adipokines (particularly leptin and adiponectin), growth factors, inflammatory markers, chemokines, metabolic and metabolomic markers could be useful to predict the existence of insulin resistance in children with obesity prior to overt glucose metabolism impairment.

Reviewer 3 Report

Review about metabolomics in obese children is clearly written and structured.

Table 1 gives a summary of the bibliography and the state of the art in the field.

The only point that should be improved is the fact that sometimes authors centers too much in their own work, as for example, when they only mention their own database for metabolomics , line 276.

Author Response

Review about metabolomics in obese children is clearly written and structured.

Table 1 gives a summary of the bibliography and the state of the art in the field.

We thank the reviewer for the consideration.

The only point that should be improved is the fact that sometimes authors centers too much in their own work, as for example, when they only mention their own database for metabolomics , line 276.

As indicated in the response to Reviewer #2, we have removed part of our self-citations and added additional citations of other authors.

However, regarding the reference about CEU Mass Mediator, apparently this was not clear as it is not a database. It is a mediator that offers the possibility to do a simultaneous query in different databases.